# RETHINKING SOFTMAX CROSS-ENTROPY LOSS FOR ADVERSARIAL ROBUSTNESS

**Tianyu Pang, Kun Xu, Yinpeng Dong, Chao Du, Ning Chen, Jun Zhu**[*]

Dept. of Comp. Sci. & Tech., BNRist Center, Institute for AI, Tsinghua University; RealAI

{pty17,xu-k16,dyp17,du-c14}@mails.tsinghua.edu.cn, {ningchen,dcszj}@tsinghua.edu.cn

## ABSTRACT

Previous work shows that adversarially robust generalization requires larger sample complexity, and the same dataset, e.g., CIFAR-10, which enables good standard accuracy may not suffice to train robust models. Since collecting new training data could be costly, we focus on better utilizing the given data by inducing the regions with high sample density in the feature space, which could lead to locally sufficient samples for robust learning. We first formally show that the softmax cross-entropy (SCE) loss and its variants convey inappropriate supervisory signals, which encourage the learned feature points to spread over the space sparsely in training. This inspires us to propose the Max-Mahalanobis center (MMC) loss to explicitly induce dense feature regions in order to benefit robustness. Namely, the MMC loss encourages the model to concentrate on learning ordered and compact representations, which gather around the preset optimal centers for different classes. We empirically demonstrate that applying the MMC loss can significantly improve robustness even under strong adaptive attacks, while keeping high accuracy on clean inputs comparable to the SCE loss with little extra computation.

## 1 INTRODUCTION

The deep neural networks (DNNs) trained by the softmax cross-entropy (SCE) loss have achieved state-of-the-art performance on various tasks (Goodfellow et al., 2016). However, in terms of robustness, the SCE loss is not sufficient to lead to satisfactory performance of the trained models. It has been widely recognized that the DNNs trained by the SCE loss are vulnerable to adversarial attacks (Carlini & Wagner, 2017a; Goodfellow et al., 2015; Kurakin et al., 2017; Moosavi-Dezfooli et al., 2016; Papernot et al., 2016), where human imperceptible perturbations can be crafted to fool a high-performance network. To improve adversarial robustness of classifiers, various kinds of defenses have been proposed, but many of them are quickly shown to be ineffective to the *adaptive attacks*, which are adapted to the specific details of the proposed defenses (Athalye et al., 2018).

Besides, the methods on verification and training provably robust networks have been proposed (Dvijotham et al., 2018a;b; Hein & Andriushchenko, 2017; Wong & Kolter, 2018). While these methods are exciting, the verification process is often slow and not scalable. Among the previously proposed defenses, the adversarial training (AT) methods can achieve state-of-the-art robustness under different adversarial settings (Madry et al., 2018; Zhang et al., 2019b). These methods either directly impose the AT mechanism on the SCE loss or add additional regularizers. Although the AT methods are relatively strong, they could sacrifice accuracy on clean inputs and are computationally expensive (Xie et al., 2019). Due to the computational obstruction, many recent efforts have been devoted to proposing faster verification methods (Wong et al., 2018; Xiao et al., 2019) and accelerating AT procedures (Shafahi et al., 2019; Zhang et al., 2019a). However, the problem still remains.

Schmidt et al. (2018) show that the sample complexity of robust learning can be significantly larger than that of standard learning. Given the difficulty of training robust classifiers in practice, they further postulate that the difficulty could stem from the insufficiency of training samples in the commonly used datasets, e.g., CIFAR-10 (Krizhevsky & Hinton, 2009). Recent work intends to solve this problem by utilizing extra unlabeled data (Carmon et al., 2019; Stanforth et al., 2019),

---

[*]Corresponding author.

while we focus on the complementary strategy to exploit the labeled data in hand more efficiently. Note that although the samples in the input space are unchangeable, we could instead manipulate the local sample distribution, i.e., sample density in the feature space via appropriate training objectives. Intuitively, by inducing high-density feature regions, there would be locally sufficient samples to train robust classifiers and return reliable predictions (Schmidt et al., 2018).

Similar to our attempt to induce high-density regions in the feature space, previous work has been proposed to improve intra-class compactness. Contrastive loss (Sun et al., 2014) and triplet loss (Schroff et al., 2015) are two classical objectives for this purpose, but the training iterations will dramatically grow to construct image pairs or triplets, which results in slow convergence and instability. The center loss (Wen et al., 2016) avoids the pair-wise or triplet-wise computation by minimizing the squared distance

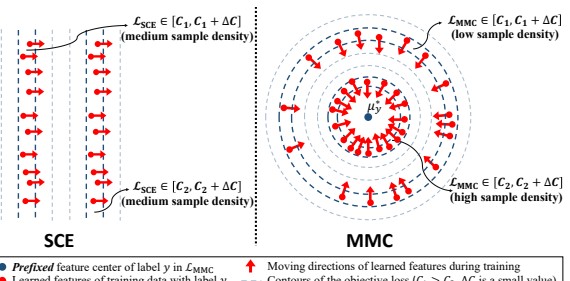

Figure 1: Intuitive illusion of how training data moves and how sample density varies in a two-dimensional feature space during the training procedure.

between the features and the corresponding class centers. However, since the class centers are updated w.r.t. the learned features during training, the center loss has to be jointly used with the SCE loss to seek for a trade-off between inter-class dispersion and intra-class compactness. Therefore, the center loss cannot concentrate on inducing strong intra-class compactness to construct high-density regions and consequently could not lead to reliable robustness, as shown in our experiments.

In this paper, we first formally analyze the sample density distribution induced by the SCE loss and its other variants (Pang et al., 2018; Wan et al., 2018) in Sec. 3.2, which demonstrates that these previously proposed objectives convey unexpected supervisory signals on the training points, which make the learned features tend to spread over the space sparsely. *This undesirable behavior mainly roots from applying the softmax function in training*, which makes the loss function only depend on the relative relation among logits and cannot directly supervise on the learned representations.

We further propose a novel training objective which can explicitly induce high-density regions in the feature space and learn more structured representations. To achieve this, we propose the **Max-Mahalanobis center (MMC) loss** (detailed in Eq. (8)) as the substitute of the SCE loss. Specifically, in the MMC loss, we first preset untrainable class centers with optimal inter-class dispersion in the feature space according to Pang et al. (2018), then we encourage the features to gather around the centers by minimizing the squared distance similar with the center loss. The MMC loss can explicitly control the inter-class dispersion by a single hyperparameter, and further concentrate on improving intra-class compactness in the training procedure to induce high-density regions, as intuitively shown in Fig. 1. Behind the simple formula, the MMC loss elegantly combines the favorable merits of the previous methods, which leads to a considerable improvement on the adversarial robustness.

In experiments, we follow the suggestion by Carlini et al. (2019) that we test under different threat models and attacks, including the *adaptive attacks* (Athalye et al., 2018) on MNIST, CIFAR-10, and CIFAR-100 (Krizhevsky & Hinton, 2009; LeCun et al., 1998). The results demonstrate that our method can lead to reliable robustness of the trained models with little extra computation, while maintaining high clean accuracy with faster convergence rates compared to the SCE loss and its variants. When combined with the existing defense mechanisms, e.g., the AT methods (Madry et al., 2018), the trained models can be further enhanced under the attacks different from the one used to craft adversarial examples for training.

## 2 PRELIMINARIES

This section first provides the notations, then introduces the adversarial attacks and threat models.

### 2.1 NOTATIONS

In this paper, we use the lowercases to denote variables and the uppercases to denote mappings. Let $L$ be the number of classes, we define the softmax function $\text{softmax}(h) : \mathbb{R}^L \rightarrow \mathbb{R}^L$ as $\text{softmax}(h)_i = \exp(h_i)/\sum_{l=1}^{L} \exp(h_l), i \in [L]$, where $[L] := \{1, \cdots, L\}$ and $h$ is termed as logit.

A deep neural network (DNN) learns a non-linear mapping from the input $x \in \mathbb{R}^p$ to the feature $z = Z(x) \in \mathbb{R}^d$. One common training objective for DNNs is the softmax cross-entropy (SCE) loss:

$$\mathcal{L}_{\text{SCE}}(Z(x), y) = -1_y^\top \log \left[ \text{softmax}(Wz + b) \right], \tag{1}$$

for a single input-label pair $(x, y)$, where $1_y$ is the one-hot encoding of $y$ and the logarithm is defined as element-wise. Here $W$ and $b$ are the weight matrix and bias vector of the SCE loss, respectively.

## 2.2 ADVERSARIAL ATTACKS AND THREAT MODELS

Previous work has shown that adversarial examples can be easily crafted to fool DNNs (Biggio et al., 2013; Nguyen et al., 2015; Szegedy et al., 2014). A large amount of attacking methods on generating adversarial examples have been introduced in recent years (Carlini & Wagner, 2017a; Chen et al., 2017; Dong et al., 2018; Goodfellow et al., 2015; Ilyas et al., 2018; Kurakin et al., 2017; Madry et al., 2018; Moosavi-Dezfooli et al., 2016; Papernot et al., 2016; Uesato et al., 2018). Given the space limit, we try to perform a comprehensive evaluation by considering five different threat models and choosing representative attacks for each threat model following Carlini et al. (2019):

**White-box $l_\infty$ distortion attack:** We apply the projected gradient descent (**PGD**) (Madry et al., 2018) method, which is efficient and widely studied in previous work (Pang et al., 2019).

**White-box $l_2$ distortion attack:** We apply the **C&W** (Carlini & Wagner, 2017a) method, which has a binary search mechanism on its parameters to find the minimal $l_2$ distortion for a successful attack.

**Black-box transfer-based attack:** We use the momentum iterative method (**MIM**) (Dong et al., 2018) that is effective on boosting adversarial transferability (Kurakin et al., 2018).

**Black-box gradient-free attack:** We choose **SPSA** (Uesato et al., 2018) since it has broken many previously proposed defenses. It can still perform well even when the loss is difficult to optimize.

**General-purpose attack:** We also evaluate the general robustness of models when adding Gaussian noise (Gilmer et al., 2019) or random rotation (Engstrom et al., 2019) on the input images.

Furthermore, to exclude the false robustness caused by, e.g., gradient mask (Athalye et al., 2018), we modify the above attacking methods to be *adaptive attacks* (Carlini & Wagner, 2017b; Carlini et al., 2019; Herley & Van Oorschot, 2017) when evaluating on the robustness of our method. The adaptive attacks are much more powerful than the non-adaptive ones, as detailed in Sec. 4.2.

## 3 METHODOLOGY

Various theoretical explanations have been developed for adversarial examples (Fawzi et al., 2016; 2018; Ilyas et al., 2019; Papernot et al., 2018). In particular, Schmidt et al. (2018) show that training robust classifiers requires significantly larger sample complexity compared to that of training standard ones, and they further postulate that the difficulty of training robust classifiers stems from, at least partly, the insufficiency of training samples in the common datasets. Recent efforts propose alternatives to benefit training with extra unlabeled data (Carmon et al., 2019; Stanforth et al., 2019), while we explore the complementary way to better use the labeled training samples for robust learning.

Although a given sample is fixed in the input space, we can instead manipulate the local sample distribution, i.e., sample density in the feature space, via designing appropriate training objectives. Intuitively, by inducing high-density regions in the feature space, it can be expected to have locally sufficient samples to train robust models that are able to return reliable predictions. In this section, we first formally define the notion of sample density in the feature space. Then we provide theoretical analyses of the sample density induced by the SCE loss and its variants. Finally, we propose our new Max-Mahalanobis center (MMC) loss and demonstrate its superiority compared to previous losses.

## 3.1 SAMPLE DENSITY IN THE FEATURE SPACE

Given a training dataset $\mathcal{D}$ with $N$ input-label pairs, and the feature mapping $Z$ trained by the objective $\mathcal{L}(Z(x), y)$ on this dataset, we define the sample density nearby the feature point $z = Z(x)$

following the similar definition in physics (Jackson, 1999) as

$$\mathbb{SD}(z) = \frac{\Delta N}{\text{Vol}(\Delta B)}. \tag{2}$$

Here $\text{Vol}(\cdot)$ denotes the volume of the input set, $\Delta B$ is a small neighbourhood containing the feature point $z$, and $\Delta N = |Z(\mathcal{D}) \cap \Delta B|$ is the number of training points in $\Delta B$, where $Z(\mathcal{D})$ is the set of all mapped features for the inputs in $\mathcal{D}$. Note that the mapped feature $z$ is still of the label $y$.

In the training procedure, the feature distribution is directly induced by the training loss $\mathcal{L}$, where minimizing the loss value is the only supervisory signal for the feature points to move (Goodfellow et al., 2016). This means that the sample density varies mainly along the orthogonal direction w.r.t. the loss contours, while the density along a certain contour could be approximately considered as the same. For example, in the right panel of Fig. 1, the sample density induced by our MMC loss (detailed in Sec. 3.3) changes mainly along the radial direction, i.e., the directions of red arrows, where the loss contours are dashed concentric circles. Therefore, supposing $\mathcal{L}(z, y) = C$, we choose $\Delta B = \{\mathbf{z} \in \mathbb{R}^d | \mathcal{L}(\mathbf{z}, y) \in [C, C + \Delta C]\}$, where $\Delta C > 0$ is a small value. Then $\text{Vol}(\Delta B)$ is the volume between the loss contours of $C$ and $C + \Delta C$ for label $y$ in the feature space.

## 3.2 THE SAMPLE DENSITY INDUCED BY THE GENERALIZED SCE LOSS

**Generalized SCE loss.** To better understand how the SCE loss and its variants (Pang et al., 2018; Wan et al., 2018) affect the sample density of features, we first generalize the definition in Eq. (1) as:

$$\mathcal{L}_{\text{g-SCE}}(Z(x), y) = -1_y^\top \log [\text{softmax}(h)], \tag{3}$$

where the logit $h = H(z) \in \mathbb{R}^L$ is a general transformation of the feature $z$, for example, $h = Wz + b$ in the SCE loss. We call this family of losses as the generalized SCE (g-SCE) loss. Wan et al. (2018) propose the large-margin Gaussian Mixture (L-GM) loss, where $h_i = -(z - \mu_i)^\top \Sigma_i (z - \mu_i) - m\delta_{i,y}$ under the assumption that the learned features $z$ distribute as a mixture of Gaussian. Here $\mu_i$ and $\Sigma_i$ are extra trainable means and covariance matrices respectively, $m$ is the margin, and $\delta_{i,y}$ is the indicator function. Pang et al. (2018) propose the Max-Mahalanobis linear discriminant analysis (MMLDA) loss, where $h_i = -\|z - \mu_i^*\|_2^2$ under the similar mixture of Gaussian assumption, but the main difference is that $\mu_i^*$ are not trainable, but calculated before training with optimal inter-class dispersion. These two losses both fall into the family of the g-SCE loss with quadratic logits:

$$h_i = -(z - \mu_i)^\top \Sigma_i (z - \mu_i) + B_i, \tag{4}$$

where $B_i$ are the bias variables. Besides, note that for the SCE loss, there is

$$\text{softmax}(Wz + b)_i = \frac{\exp(W_i^\top z + b_i)}{\sum_{l \in [L]} \exp(W_l^\top z + b_l)} = \frac{\exp(-\|z - \frac{1}{2}W_i\|_2^2 + b_i + \frac{1}{4}\|W_i\|_2^2)}{\sum_{l \in [L]} \exp(-\|z - \frac{1}{2}W_l\|_2^2 + b_l + \frac{1}{4}\|W_l\|_2^2)}.$$

According to Eq. (4), the SCE loss can also be regraded as a special case of the g-SCE loss with quadratic logits, where $\mu_i = \frac{1}{2}W_i$, $B_i = b_i + \frac{1}{4}\|W_i\|_2^2$ and $\Sigma_i = I$ are identity matrices. Therefore, later when we refer to the g-SCE loss, we assume that the logits are quadratic as in Eq. (4) by default.

**The contours of the g-SCE loss.** To provide a formal representation of the sample density induced by the g-SCE loss, we first derive the formula of the contours, i.e., the closed-form solution of $\mathcal{L}_{\text{g-SCE}}(Z(x), y) = C$ in the space of $z$, where $C \in (0, +\infty)$ is a given constant. Let $C_e = \exp(C) \in (1, +\infty)$, from Eq. (3), we can represent the contours as the solution of

$$\log \left( 1 + \frac{\sum_{l \neq y} \exp(h_l)}{\exp(h_y)} \right) = C \implies h_y = \log \left[ \sum_{l \neq y} \exp(h_l) \right] - \log(C_e - 1). \tag{5}$$

The function in Eq. (5) does not provide an intuitive closed-form solution for the contours, since the existence of the term $\log \left[ \sum_{l \neq y} \exp(h_l) \right]$. However, note that this term belongs to the family of Log-Sum-Exp (LSE) function, which is a smooth approximation to the maximum function (Nesterov, 2005; Nielsen & Sun, 2016). Therefore, we can locally approximate the function in Eq. (5) with

$$h_y - h_{\tilde{y}} = -\log(C_e - 1), \tag{6}$$

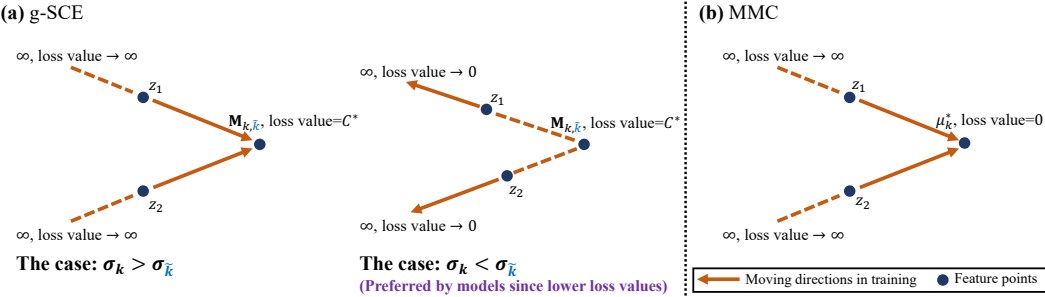

Figure 2: Intuitive illustration on the inherent limitations of the g-SCE loss. Reasonably learned features for a classification task should distribute in clusters, so it is counter-intuitive that the feature points tend to move to infinity to pursue lower loss values when applying the g-SCE loss. In contrast, MMC induces models to learn more structured and orderly features.

where $\tilde{y} = \arg\max_{l \neq y} h_l$. In the following text, we apply colored characters with tilde like $\tilde{y}$ to better visually distinguish them. According to Eq. (6), we can define $\mathcal{L}_{y,\tilde{y}}(z) = \log[\exp(h_{\tilde{y}} - h_y) + 1]$ as the local approximation of the g-SCE loss nearby the feature point $z$, and substitute the neighborhood $\Delta B$ by $\Delta B_{y,\tilde{y}} = \{\mathbf{z} \in \mathbb{R}^d | \mathcal{L}_{y,\tilde{y}}(\mathbf{z}) \in [C, C + \Delta C]\}$. For simplicity, we assume scaled identity covariance matrix in Eq. (4), i.e., $\Sigma_i = \sigma_i I$, where $\sigma_i > 0$ are scalars. Through simple derivations (detailed in Appendix A.1), we show that if $\sigma_y \neq \sigma_{\tilde{y}}$, the solution of $\mathcal{L}_{y,\tilde{y}}(z) = C$ is a $(d-1)$-dimensional hypersphere with the center $\mathbf{M}_{y,\tilde{y}} = (\sigma_y - \sigma_{\tilde{y}})^{-1}(\sigma_y \mu_y - \sigma_{\tilde{y}} \mu_{\tilde{y}})$; otherwise if $\sigma_y = \sigma_{\tilde{y}}$, the hypersphere-shape contour will degenerate to a hyperplane.

**The induced sample density.** Since the approximation in Eq. (6) depends on the specific $y$ and $\tilde{y}$, we define the training subset $\mathcal{D}_{k,\tilde{k}} = \{(x, y) \in \mathcal{D} | y = k,\ \tilde{y} = \tilde{k}\}$ and $N_{k,\tilde{k}} = |\mathcal{D}_{k,\tilde{k}}|$. Intuitively, $\mathcal{D}_{k,\tilde{k}}$ includes the data with the true label of class $k$, while the highest prediction returned by the classifier is class $\tilde{k}$ among other classes. Then we can derive the approximated sample density in the feature space induced by the g-SCE loss, as stated in the following theorem:

**Theorem 1.** *(Proof in Appendix A.1) Given* $(x, y) \in \mathcal{D}_{k,\tilde{k}}$, $z = Z(x)$ *and* $\mathcal{L}_{g\text{-}SCE}(z, y) = C$, *if there are* $\Sigma_k = \sigma_k I$, $\Sigma_{\tilde{k}} = \sigma_{\tilde{k}} I$, *and* $\sigma_k \neq \sigma_{\tilde{k}}$, *then the sample density nearby the feature point $z$ based on the approximation in Eq. (6) is*

$$\mathbb{SD}(z) \propto \frac{N_{k,\tilde{k}} \cdot p_{k,\tilde{k}}(C)}{\left[\mathbf{B}_{k,\tilde{k}} + \frac{\log(C_e - 1)}{\sigma_k - \sigma_{\tilde{k}}}\right]^{\frac{d-1}{2}}}, \text{ and } \mathbf{B}_{k,\tilde{k}} = \frac{\sigma_k \sigma_{\tilde{k}} \|\mu_k - \mu_{\tilde{k}}\|_2^2}{(\sigma_k - \sigma_{\tilde{k}})^2} + \frac{B_k - B_{\tilde{k}}}{\sigma_k - \sigma_{\tilde{k}}}, \quad (7)$$

*where for the input-label pair in* $\mathcal{D}_{k,\tilde{k}}$, *there is* $\mathcal{L}_{g\text{-}SCE} \sim p_{k,\tilde{k}}(c)$.

**Limitations of the g-SCE loss.** Based on Theorem 1 and the approximation in Eq. (6), let $C^* = \log(1 + \exp(\mathbf{B}_{k,\tilde{k}}(\sigma_{\tilde{k}} - \sigma_k)))$ and $C_e^* = \exp(C^*)$, such that $\mathbf{B}_{k,\tilde{k}} + \frac{\log(C_e^* - 1)}{\sigma_k - \sigma_{\tilde{k}}} = 0$. According to Appendix A.1, if $\sigma_k > \sigma_{\tilde{k}}$, then $C^*$ will act as a tight lower bound for $C$, i.e., the solution set of $C < C^*$ is empty. This will make the training procedure tend to avoid this case since the loss $C$ cannot be further minimized to zero, which will introduce unnecessary biases on the returned predictions. On the other hand, if $\sigma_k < \sigma_{\tilde{k}}$, $C$ could be minimized to zero. However, when $C \rightarrow 0$, the sample density will also tend to zero since there is $\mathbf{B}_{k,\tilde{k}} + \frac{\log(C_e - 1)}{\sigma_k - \sigma_{\tilde{k}}} \rightarrow \infty$, which means the feature point will be encouraged to go further and further from the hypersphere center $\mathbf{M}_{k,\tilde{k}}$ only to make the loss value $C$ be lower, as intuitively illustrated in Fig. 2(a).

This counter-intuitive behavior mainly roots from applying the softmax function in training. Namely, the softmax normalization makes the loss value only depend on the relative relation among logits. This causes indirect and unexpected supervisory signals on the learned features, such that the points with low loss values tend to spread over the space sparsely. Fortunately, in practice, the feature point will not really move to infinity, since the existence of batch normalization layers (Ioffe & Szegedy, 2015), and the squared radius from the center $\mathbf{M}_{k,\tilde{k}}$ increases as $\mathcal{O}(|\log C|)$ when minimizing the loss $C$. These theoretical conclusions are consistent with the empirical observations on the two-dimensional features in previous work (*cf.* Fig. 1 in Wan et al. (2018)).

Another limitation of the g-SCE loss is that the sample density is proportional to $N_{k,\tilde{k}}$, which is on average $N/L^2$. For example, there are around $1.3$ million training data in ImageNet (Deng et al., 2009), but with a large number of classes $L = 1,000$, there are averagely less than two samples in each $\mathcal{D}_{k,\tilde{k}}$. These limitations inspire us to design the new training loss as in Sec 3.3.

**Remark 1.** If $\sigma_k = \sigma_{\tilde{k}}$ (e.g., as in the SCE loss), the features with loss values in $[C, C + \Delta C]$ will be encouraged to locate between two hyperplane contours without further supervision, and consequently there will not be explicit supervision on the sample density as shown in the left panel of Fig. 1.

**Remark 2.** Except for the g-SCE loss, Wen et al. (2016) propose the center loss in order to improve the intra-class compactness of learned features, formulated as $\mathcal{L}_{\text{Center}}(Z(x), y) = \frac{1}{2}\|z - \mu_y\|_2^2$. Here the center $\mu_y$ is updated based on a mini-batch of learned features with label $y$ in each training iteration. The center loss has to be jointly used with the SCE loss as $\mathcal{L}_{\text{SCE}} + \lambda\mathcal{L}_{\text{Center}}$, since simply supervise the DNNs with the center loss term will cause the learned features and centers to degrade to zeros (Wen et al., 2016). This makes it difficult to derive a closed-form formula for the induced sample density. Besides, the center loss method cannot concentrate on improving intra-class compactness, since it has to seek for a trade-off between inter-class dispersion and intra-class compactness.

## 3.3 MAX-MAHALANOBIS CENTER LOSS

Inspired by the above analyses, we propose the **Max-Mahalanobis center (MMC) loss** to explicitly learn more structured representations and induce high-density regions in the feature space. The MMC loss is defined in a regression form without the softmax function as

$$\mathcal{L}_{\text{MMC}}(Z(x), y) = \frac{1}{2}\|z - \mu_y^*\|_2^2. \tag{8}$$

Here $\mu^* = \{\mu_l^*\}_{l\in[L]}$ are the centers of the Max-Mahalanobis distribution (MMD) (Pang et al., 2018). The MMD is a mixture of Gaussian distribution with identity covariance matrix and preset centers $\mu^*$, where $\|\mu_l^*\|_2 = C_{\text{MM}}$ for any $l \in [L]$, and $C_{\text{MM}}$ is a hyperparameter. These MMD centers are invariable during training, which are crafted according to the criterion: $\mu^* = \arg\min_\mu \max_{i\neq j}\langle\mu_i, \mu_j\rangle$. Intuitively, this criterion is to maximize the minimal angle between any two centers, which can provide optimal inter-class dispersion as shown in Pang et al. (2018). In Appendix B.1, we provide the generation algorithm for $\mu^*$ in MMC. We derive the sample density induced by the MMC loss in the feature space, as stated in Theorem 2. Similar to the previously introduced notations, here we define the subset $\mathcal{D}_k = \{(x, y) \in \mathcal{D}|y = k\}$ and $N_k = |\mathcal{D}_k|$.

**Theorem 2.** *(Proof in Appendix A.2) Given $(x, y) \in \mathcal{D}_k$, $z = Z(x)$ and $\mathcal{L}_{MMC}(z, y) = C$, the sample density nearby the feature point $z$ is*

$$\mathbb{SD}(z) \propto \frac{N_k \cdot p_k(C)}{C^{\frac{d-1}{2}}}, \tag{9}$$

*where for the input-label pair in $\mathcal{D}_k$, there is $\mathcal{L}_{MMC} \sim p_k(c)$.*

According to Theorem 2, there are several attractive merits of the MMC loss, as described below.

**Inducing higher sample density.** Compared to Theorem 1, the sample density induced by MMC is proportional to $N_k$ rather than $N_{k,\tilde{k}}$, where $N_k$ is on average $N/L$. It facilitates producing higher sample density. Furthermore, when the loss value $C$ is minimized to zero, the sample density will exponentially increase according to Eq. (9), as illustrated in Fig. 2(b). The right panel of Fig. 1 also provides an intuitive insight on this property of the MMC loss: since the loss value $C$ is proportional to the squared distance from the preset center $\mu_y^*$, the feature points with lower loss values are certain to locate in a smaller volume around the center. Consequently, the feature points of the same class are encouraged to gather around the corresponding center, such that for each sample, there will be locally enough data in its neighborhood for robust learning (Schmidt et al., 2018). The MMC loss value also becomes a reliable metric of the uncertainty on returned predictions.

**Better exploiting model capacity.** Behind the simple formula, the MMC loss can explicitly monitor inter-class dispersion by the hyperparameter $C_{\text{MM}}$, while enabling the network to concentrate on minimizing intra-class compactness in training. Instead of repeatedly searching for an internal trade-off in training as the center loss, the monotonicity of the supervisory signals induced by MMC can better exploit model capacity and also lead to faster convergence, as empirically shown in Fig. 3(a).

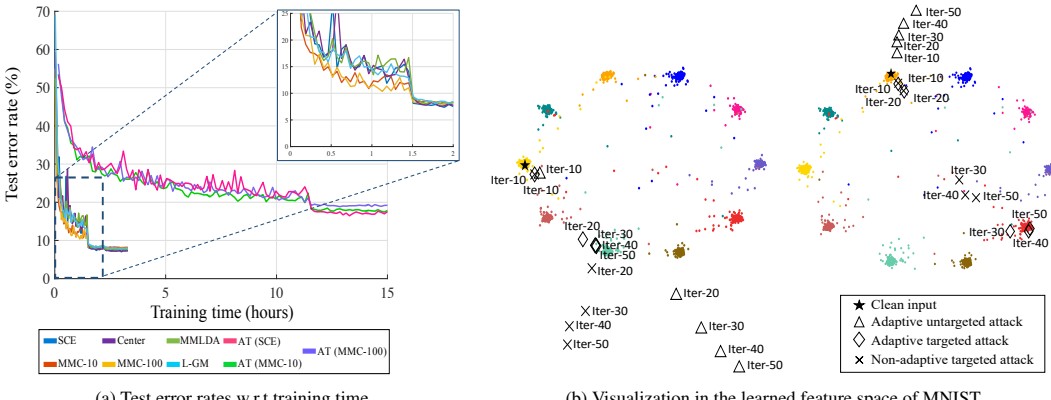

(a) Test error rates w.r.t training time

(b) Visualization in the learned feature space of MNIST

Figure 3: **(a)** Test error rates on clean images w.r.t training time on **CIFAR-10**. Here AT refers to 10-steps targeted PGD adversarial training, i.e., AT$_{10}^{\textbf{tar}}$. **(b)** Two-dimensional visualization of the attacks on trained MMC networks in the feature space of MNIST. For each attack there is $\epsilon = 0.3$ with step size of $0.01$. The total number of iteration steps is 50, where Iter-$n$ indicates the perturbed features at $n$-th iteration step.

**Avoiding the degradation problem.** The MMC loss can naturally avoid the degradation problem encountered in Wen et al. (2016) when the center loss is not jointly used with the SCE loss, since the preset centers $\mu^*$ for MMC are untrainable. In the test phase, the network trained by MMC can still return a normalized prediction with the softmax function. More details about the empirical superiorities of the MMC loss over other previous losses are demonstrated in Sec. 4.

**Remark 3.** In Appendix B.2, we discuss on why the squared-error form in Eq. (8) is preferred compared to, e.g., the absolute form or the Huber form in the adversarial setting. We further introduce flexible variants of the MMC loss in Appendix B.3, which can better adapt to various tasks.

**Remark 4.** Pang et al. (2018) propose a Max-Mahalanobis linear discriminant analysis (MMLDA) method, which assumes the features to distribute as an MMD. Due to the Gaussian mixture assumption, the training loss for the MMLDA method is obtained by the Bayes' theorem as

$$\mathcal{L}_{\text{MMLDA}}(Z(x), y) = -\log\left[\frac{\exp(-\frac{\|z - \mu_y^*\|_2^2}{2})}{\sum_{l \in [L]} \exp(-\frac{\|z - \mu_l^*\|_2^2}{2})}\right] = -\log\left[\frac{\exp(z^\top \mu_y^*)}{\sum_{l \in [L]} \exp(z^\top \mu_l^*)}\right]. \quad (10)$$

Note that there is $\Sigma_i = \frac{1}{2}I$ in Eq. (4) for the MMLDA loss, similar with the SCE loss. Thus the MMLDA method cannot explicitly supervise on the sample density and induce high-density regions in the feature space, as analyzed in Sec. 3.2. Compared to the MMLDA method, the MMC loss introduces extra supervision on intra-class compactness, which facilitates better robustness.

## 4 EXPERIMENTS

In this section, we empirically demonstrate several attractive merits of applying the MMC loss. We experiment on the widely used MNIST, CIFAR-10, and CIFAR-100 datasets (Krizhevsky & Hinton, 2009; LeCun et al., 1998). The main baselines for the MMC loss are SCE (He et al., 2016), Center loss (Wen et al., 2016), MMLDA (Pang et al., 2018), and L-GM (Wan et al., 2018). The codes are provided in https://github.com/P2333/Max-Mahalanobis-Training.

### 4.1 PERFORMANCE ON THE CLEAN INPUTS

The network architecture applied is ResNet-32 with five core layer blocks (He et al., 2016). Here we use MMC-10 to indicate the MMC loss with $C_{\text{MM}} = 10$, where $C_{\text{MM}}$ is assigned based on the cross-validation results in Pang et al. (2018). The hyperparameters for the center loss, L-GM loss and the MMLDA method all follow the settings in the original papers (Pang et al., 2018; Wan et al., 2018; Wen et al., 2016). The pixel values are scaled to the interval $[0, 1]$. For each training loss with or without the AT mechanism, we apply the momentum SGD (Qian, 1999) optimizer with the initial learning rate of $0.01$, and train for 40 epochs on MNIST, 200 epochs on CIFAR-10 and CIFAR-100. The learning rate decays with a factor of $0.1$ at $100$ and $150$ epochs, respectively.

Table 1: Classification accuracy (%) on the *white-box* adversarial examples crafted on the test set of **CIFAR-10**. The superscript **tar** indicates targeted attacks, while **un** indicates untargeted attacks. The subscripts indicate the number of iteration steps when performing attacks. The results w.r.t the MMC loss are reported under the adaptive versions of different attacks. The notation $\leq 1$ represents accuracy less than 1%. The MMC-10 (rand) is an ablation study where the class centers are uniformly sampled on the hypersphere.

| Methods | Clean | Perturbation $\epsilon = 8/255$ | | | | Perturbation $\epsilon = 16/255$ | | | |
|---|---|---|---|---|---|---|---|---|---|
| | | $\text{PGD}_{10}^{\text{tar}}$ | $\text{PGD}_{10}^{\text{un}}$ | $\text{PGD}_{50}^{\text{tar}}$ | $\text{PGD}_{50}^{\text{un}}$ | $\text{PGD}_{10}^{\text{tar}}$ | $\text{PGD}_{10}^{\text{un}}$ | $\text{PGD}_{50}^{\text{tar}}$ | $\text{PGD}_{50}^{\text{un}}$ |
| SCE | 92.9 | $\leq 1$ | 3.7 | $\leq 1$ | 3.6 | $\leq 1$ | 2.9 | $\leq 1$ | 2.6 |
| Center loss | 92.8 | $\leq 1$ | 4.4 | $\leq 1$ | 4.3 | $\leq 1$ | 3.1 | $\leq 1$ | 2.9 |
| MMLDA | 92.4 | $\leq 1$ | 16.5 | $\leq 1$ | 9.7 | $\leq 1$ | 6.7 | $\leq 1$ | 5.5 |
| L-GM | 92.5 | 37.6 | 19.8 | 8.9 | 4.9 | 26.0 | 11.0 | 2.5 | 2.8 |
| **MMC-10** (rand) | 92.3 | 43.5 | 29.2 | 20.9 | 18.4 | 31.3 | 17.9 | 8.6 | 11.6 |
| **MMC-10** | 92.7 | **48.7** | **36.0** | **26.6** | **24.8** | **36.1** | **25.2** | **13.4** | **17.5** |
| $\text{AT}_{10}^{\text{tar}}$ (SCE) | 83.7 | **70.6** | 49.7 | **69.8** | 47.8 | 48.4 | 26.7 | 31.2 | 16.0 |
| $\text{AT}_{10}^{\text{tar}}$ (**MMC-10**) | 83.0 | 69.2 | **54.8** | 67.0 | **53.5** | **58.6** | **47.3** | **44.7** | **45.1** |
| $\text{AT}_{10}^{\text{un}}$ (SCE) | 80.9 | 69.8 | 55.4 | 69.4 | 53.9 | 53.3 | 34.1 | 38.5 | 21.5 |
| $\text{AT}_{10}^{\text{un}}$ (**MMC-10**) | 81.8 | **70.8** | **56.3** | **70.1** | **55.0** | **54.7** | **37.4** | **39.9** | **27.7** |

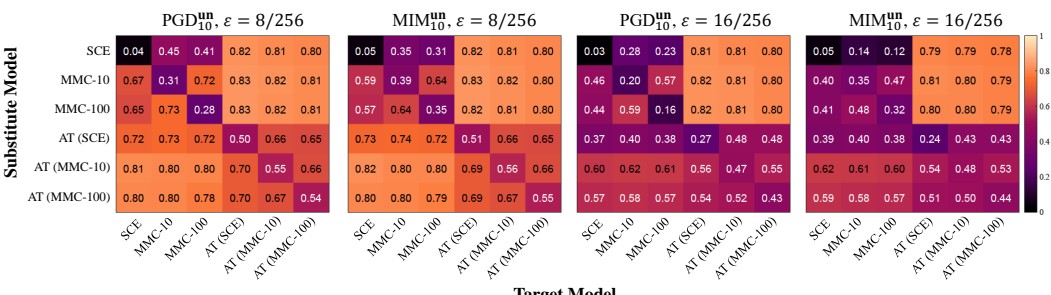

Figure 4: Classification accuracy under the black-box transfer-based attacks on the test set of **CIFAR-10**. The *substitute model* refers to the one used to craft adversarial examples, and the *target model* is the one that an adversary actually intends to fool. Here AT refers to $\text{AT}_{10}^{\text{tar}}$.

When applying the AT mechanism (Madry et al., 2018), the adversarial examples for training are crafted by 10-steps targeted or untargeted PGD with $\epsilon = 8/255$. In Fig. 3(a), we provide the curves of the test error rate w.r.t. the training time. Note that the MMC loss induces faster convergence rate and requires little extra computation compared to the SCE loss and its variants, while keeping comparable performance on the clean images. In comparison, implementing the AT mechanism is computationally expensive in training and will sacrifice the accuracy on the clean images.

## 4.2 ADAPTIVE ATTACKS FOR THE MMC LOSS

As stated in Athalye et al. (2018), only applying the existing attacks with default hyperparameters is not sufficient to claim reliable robustness. Thus, we apply the adaptive versions of existing attacks when evading the networks trained by the MMC loss (detailed in Appendix B.4). For instance, the non-adaptive objectives for PGD are variants of the SCE loss (Madry et al., 2018), while the adaptive objectives are $-\mathcal{L}_{\text{MMC}}(z, y)$ and $\mathcal{L}_{\text{MMC}}(z, y_t)$ in the untargeted and targeted modes for PGD, respectively. Here $y_t$ is the target label. To verify that the adaptive attacks are more effective than the non-adaptive ones, we modify the network architecture with a two-dimensional feature layer and visualize the PGD attacking procedure in Fig. 3(b). The two panels separately correspond to two randomly selected clean inputs indicated by black stars. The ten colored clusters in each panel consist of the features of all the 10,000 test samples in MNIST, where each color corresponds to one class. We can see that the adaptive attacks are indeed much more efficient than the non-adaptive one.

## 4.3 PERFORMANCE UNDER THE WHITE-BOX ATTACKS

We first investigate the white-box $l_\infty$ distortion setting using the PGD attack, and report the results in Table 1. According to Carlini et al. (2019), we evaluate under different combinations of the attacking parameters: the perturbation $\epsilon$, iteration steps, and the attack mode, i.e., targeted or untargeted.

Table 2: Experiments on **CIFAR-10**. **Part I:** Averaged $l_2$ distortion of the white-box adversarial examples crafted by C&W with 1,000 iteration steps. **Part II:** Classification accuracy (%) under the block-box SPSA attack. **Part III:** Classification accuracy (%) under general transformations. The standard deviation $\sigma$ for the Gaussian noise is 0.05, the degree range is $\pm 30°$ for random rotation.

| Methods | Part I | | Part II ($\epsilon=8/255$) | | Part II ($\epsilon=16/255$) | | Part III | |
|---|---|---|---|---|---|---|---|---|
| | C&W$^{\text{tar}}$ | C&W$^{\text{un}}$ | SPSA$_{10}^{\text{tar}}$ | SPSA$_{10}^{\text{un}}$ | SPSA$_{10}^{\text{tar}}$ | SPSA$_{10}^{\text{un}}$ | Noise | Rotation |
| SCE | 0.12 | 0.07 | 12.3 | 1.2 | 5.1 | $\leq 1$ | 52.0 | 83.5 |
| Center loss | 0.13 | 0.07 | 21.2 | 6.0 | 10.6 | 2.0 | 55.4 | 84.9 |
| MMLDA | 0.17 | 0.10 | 25.6 | 13.2 | 11.3 | 5.7 | 57.9 | 84.8 |
| L-GM | 0.23 | 0.12 | 61.9 | 45.9 | 46.1 | 28.2 | 59.2 | 82.4 |
| MMC-10 | **0.34** | **0.17** | **69.5** | **56.9** | **57.2** | **41.5** | **69.3** | **87.2** |
| AT$_{10}^{\text{tar}}$ (SCE) | 1.19 | 0.63 | **81.1** | 67.8 | **77.9** | 59.4 | 82.2 | **76.0** |
| AT$_{10}^{\text{tar}}$ (MMC-10) | **1.91** | **0.85** | 79.1 | **69.2** | 74.5 | **62.7** | **83.5** | 75.2 |
| AT$_{10}^{\text{un}}$ (SCE) | 1.26 | 0.68 | 78.8 | 67.0 | 73.7 | 60.3 | 78.9 | 73.7 |
| AT$_{10}^{\text{un}}$ (MMC-10) | **1.55** | **0.73** | **80.4** | **69.6** | **74.6** | **62.4** | **80.3** | **75.8** |

Following the setting in Madry et al. (2018), we choose the perturbation $\epsilon = 8/255$ and $16/255$, with the step size be $2/255$. We have also run PGD-100 and PGD-200 attacks, and find that the accuracy converges compared to PGD-50. In each PGD experiment, we ran several times with different random restarts to guarantee the reliability of the reported results.

**Ablation study.** To investigate the effect on robustness induced by high sample density in MMC, we substitute uniformly sampled center set (Liu et al., 2018; Duan et al., 2019), i.e., $\mu^r = \{\mu_l^r\}_{l \in [L]}$ for the MM center set $\mu^*$, and name the resulted method as "MMC-10 (rand)" as shown in Table 1. There is also $\|\mu_l^r\|_2 = C_{\text{MM}}$, but $\mu^r$ is no longer the solution of the min-max problem in Sec. 3.3.

From the results in Table 1, we can see that higher sample density alone in "MMC-10 (rand)" can already lead to much better robustness than other baseline methods even under the adaptive attacks, while using the optimal center set $\mu^*$ as in "MMC-10" can further improve performance. When combining with the AT mechanism, the trained models have better performance under the attacks different from the one used to craft adversarial examples for training, e.g, PGD$_{50}^{\text{un}}$ with $\epsilon = 16/255$.

Then we investigate the white-box $l_2$ distortion setting. We apply the C&W attack, where it has a binary search mechanism to find the minimal distortion to successfully mislead the classifier under the untargeted mode, or lead the classifier to predict the target label in the targeted mode. Following the suggestion in Carlini & Wagner (2017a), we set the binary search steps to be 9 with the initial constant $c = 0.01$. The iteration steps for each value of $c$ are set to be 1,000 with the learning rate of 0.005. In the Part I of Table 2, we report the minimal distortions found by the C&W attack. As expected, it requires much larger distortions to successfully evade the networks trained by MMC.

## 4.4 PERFORMANCE UNDER THE BLACK-BOX ATTACKS

As suggested in Carlini et al. (2019), providing evidence of being robust against the black-box attacks is critical to claim reliable robustness. We first perform the transfer-based attacks using PGD and MIM. Since the targeted attacks usually have poor transferability (Kurakin et al., 2018), we only focus on the untargeted mode in this case, and the results are shown in Fig. 4. We further perform the gradient-free attacks using the SPSA method and report the results in the Part II of Table 2. To perform numerical approximations on gradients in SPSA, we set the batch size to be 128, the learning rate is 0.01, and the step size of the finite difference is $\delta = 0.01$, as suggested by Uesato et al. (2018). We also evaluate under stronger

Table 3: Accuracy (%) of MMC-10 under SPSA with different batch sizes.

| CIFAR-10 | | |
|---|---|---|
| Batch | SPSA$_{10}^{\text{un}}$ | SPSA$_{10}^{\text{tar}}$ |
| 128 | 57.0 | 69.0 |
| 4096 | 41.0 | 52.0 |
| 8192 | 37.0 | 49.0 |

SPSA attacks with batch size to be 4096 and 8192 in Table 3, where the $\epsilon = 8/255$. With larger batch sizes, we can find that the accuracy under the black-box SPSA attacks converges to it under the white-box PGD attacks. These results indicate that training with the MMC loss also leads to

Table 4: Experiments on **CIFAR-100**. **Part I:** Classification accuracy (%) on the clean test samples. **Part II:** Classification accuracy (%) under the white-box PGD attacks and the block-box SPSA attack. The attacks are adaptive for MMC. Here the batch size for SPSA is 128. **Part III:** Averaged $l_2$ distortion of the white-box adversarial examples crafted by C&W with 1,000 iteration steps and 9 binary search epochs.

| | **Part I** | **Part II** ($\epsilon = 8/255$) | | | | **Part I** | |
| Methods | Clean | $\text{PGD}_{10}^{\text{tar}}$ | $\text{PGD}_{10}^{\text{un}}$ | $\text{SPSA}_{10}^{\text{tar}}$ | $\text{SPSA}_{10}^{\text{un}}$ | $\text{C\&W}^{\text{tar}}$ | $\text{C\&W}^{\text{un}}$ |
|---|---|---|---|---|---|---|---|
| SCE | 72.9 | $\leq 1$ | 8.0 | 14.0 | 1.9 | 0.16 | 0.047 |
| Center | 72.8 | $\leq 1$ | 10.2 | 14.7 | 2.3 | 0.18 | 0.048 |
| MMLDA | 72.2 | $\leq 1$ | 13.9 | 18.5 | 5.6 | 0.21 | 0.050 |
| L-GM | 71.3 | 15.8 | 15.3 | 22.8 | 7.6 | 0.31 | 0.063 |
| MMC-10 | 71.9 | **23.9** | **23.4** | **33.4** | **15.8** | **0.37** | **0.085** |

robustness under the black-box attacks, which verifies that our method can induce reliable robustness, rather than the false one caused by, e.g., gradient mask (Athalye et al., 2018).

### 4.5 PERFORMANCE UNDER THE GENERAL-PURPOSE ATTACKS

To show that our method is generally robust, we further test under the general-purpose attacks (Carlini et al., 2019). We apply the Gaussian noise (Fawzi et al., 2016; Gilmer et al., 2019) and rotation transformation (Engstrom et al., 2019), which are not included in the data augmentation for training. The results are given in the Part III of Table 2. Note that the AT methods are less robust to simple transformations like rotation, as also observed in previous work (Engstrom et al., 2019). In comparison, the models trained by the MMC loss are still robust to these easy-to-apply attacks.

### 4.6 EXPERIMENTS ON CIFAR-100

In Table 4 and Table 5, we provide the results on CIFAR-100 under the white-box PGD and C&W attacks, and the black-box gradient-free SPSA attack. The hyperparameter setting for each attack is the same as it on CIFAR-10. Compared to previous defense strategies which also evaluate on CIFAR-100 (Pang et al., 2019; Mustafa et al., 2019), MMC can improve robustness more significantly, while keeping better performance on the clean inputs. Compared to the results on CIFAR-10, the averaged distortion of C&W on CIFAR-100 is larger for a successful targeted attack and is much smaller for a successful untargeted attack. This is because when only the number of classes increases, e.g., from 10 to 100, it is easier to achieve a coarse untargeted attack, but harder to make a subtle targeted attack. Note that in Table 5, we also train on the ResNet-110 model with eighteen core block layers except for the ResNet-32 model. The results show that MMC can further benefit from deep network architectures and better exploit model capacity to improve robustness. Similar properties are also observed in previous work when applying the AT methods (Madry et al., 2018). In contrast, as shown in Table 5, the models trained by SCE are comparably sensitive to adversarial perturbations for different architectures, which demonstrates that SCE cannot take full advantage of the model capacity to improve robustness. This verifies that MMC provides effective robustness promoting mechanism like the AT methods, with much less computational cost.

## 5 CONCLUSION

In this paper, we formally demonstrate that applying the softmax function in training could potentially lead to unexpected supervisory signals. To solve this problem, we propose the MMC loss to learn more structured representations and induce high-density regions in the feature space. In our experiments, we empirically demonstrate several favorable merits of our method: **(i)** Lead to reliable robustness even under strong adaptive attacks in different threat models; **(ii)** Keep high performance on clean inputs comparable to SCE; **(iii)** Introduce little extra computation compared to the SCE loss; **(iv)** Compatible with the existing defense mechanisms, e.g., the AT methods. Our analyses in this paper also provide useful insights for future work on designing new objectives beyond the SCE framework.

## ACKNOWLEDGEMENTS

This work was supported by the National Key Research and Development Program of China (No. 2017YFA0700904), NSFC Projects (Nos. 61620106010, U19B2034, U1811461), Beijing NSF Project (No. L172037), Beijing Academy of Artificial Intelligence (BAAI), Tsinghua-Huawei Joint Research Program, a grant from Tsinghua Institute for Guo Qiang, Tiangong Institute for Intelligent Computing, the JP Morgan Faculty Research Program and the NVIDIA NVAIL Program with GPU/DGX Acceleration.

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

## A PROOF

In this section, we provide the proof of the theorems proposed in the paper.

### A.1 PROOF OF THEOREM 1

According to the definition of sample density

$$\mathbb{SD}(z) = \frac{\Delta N}{\text{Vol}(\Delta B)},$$

we separately calculate $\Delta N$ and $\text{Vol}(\Delta B)$. Since $\mathcal{L}_{\text{g-SCE}} \sim p_{k,\tilde{k}}(c)$ for the data points in $\mathcal{D}_{k,\tilde{k}}$, recall that $\Delta B = \{z \in \mathbb{R}^d | \mathcal{L}_{\text{g-SCE}} \in [C, C + \Delta C]\}$, then there is

$$\begin{aligned} \Delta N &= |Z(\mathcal{D}_{k,\tilde{k}}) \cap \Delta B| \\ &= N_{k,\tilde{k}} \cdot p_{k,\tilde{k}}(C) \cdot \Delta C. \end{aligned} \quad (11)$$

Now we calculate $\text{Vol}(\Delta B)$ by approximating it with $\text{Vol}(\Delta B_{y,\tilde{y}})$. We first derive the solution of $\mathcal{L}_{y,\tilde{y}} = C$. For simplicity, we assume scaled identity covariance matrix, i.e., $\Sigma_i = \sigma_i I$, where $\sigma_i > 0$ are scalars. Then $\forall i, j \in [L]$, $c$ is any constant, if $\sigma_i \neq \sigma_j$, the solution of $h_i - h_j = c$ is a $(d-1)$-dimensional hypersphere embedded in the $d$-dimensional space of the feature $z$:

$$\|z - \mathbf{M}_{i,j}\|_2^2 = \mathbf{B}_{i,j} - \frac{c}{\sigma_i - \sigma_j}, \text{ where } \mathbf{M}_{i,j} = \frac{\sigma_i \mu_i - \sigma_j \mu_j}{\sigma_i - \sigma_j}, \ \mathbf{B}_{i,j} = \frac{\sigma_i \sigma_j \|\mu_i - \mu_j\|_2^2}{(\sigma_i - \sigma_j)^2} + \frac{B_i - B_j}{\sigma_i - \sigma_j}. \quad (12)$$

Note that each value of $c$ corresponds to a specific contour, where $\mathbf{M}_{i,j}$ and $\mathbf{B}_{i,j}$ can be regraded as constant w.r.t. $c$. When $\mathbf{B}_{i,j} < (\sigma_i - \sigma_j)^{-1}c$, the solution set becomes empty. Specially, if $\sigma_i = \sigma_j = \sigma$, the hypersphere-shape contour will degenerate to a hyperplane: $z^\top(\mu_i - \mu_j) = \frac{1}{2}\left[\|\mu_i\|_2^2 - \|\mu_j\|_2^2 + \sigma^{-1}(B_j - B_i + c)\right]$. For example, for the SCE loss, the solution of the contour is $z^\top(W_i - W_j) = b_j - b_i + c$. For more general $\Sigma_i$, the conclusions are similar, e.g., the solution in Eq. (12) will become a hyperellipse. Now it easy to show that the solution of $\mathcal{L}_{y,\tilde{y}} = C$ when $y = k, \tilde{y} = \tilde{k}$ is the hypersphere:

$$\|z - \mathbf{M}_{k,\tilde{k}}\|_2^2 = \mathbf{B}_{k,\tilde{k}} + \frac{\log(C_e - 1)}{\sigma_k - \sigma_{\tilde{k}}}. \quad (13)$$

According to the formula of the hypersphere surface area (Loskot & Beaulieu, 2007), the volume of $\Delta B_{y,\tilde{y}}$ is

$$\text{Vol}(\Delta B_{y,\tilde{y}}) = \frac{2\pi^{\frac{d}{2}}}{\Gamma(\frac{d}{2})} \left(\mathbf{B}_{k,\tilde{k}} + \frac{\log(C_e - 1)}{\sigma_k - \sigma_{\tilde{k}}}\right)^{\frac{d-1}{2}} \cdot \Delta C, \quad (14)$$

where $\Gamma(\cdot)$ is the gamma function. Finally we can approximate the sample density as

$$\begin{aligned} \mathbb{SD}(z) &\approx \frac{\Delta N}{\Delta B_{y,\tilde{y}}} \\ &\propto \frac{N_{k,\tilde{k}} \cdot p_{k,\tilde{k}}(C)}{\left[\mathbf{B}_{k,\tilde{k}} + \frac{\log(C_e-1)}{\sigma_k - \sigma_{\tilde{k}}}\right]^{\frac{d-1}{2}}}. \end{aligned} \quad (15)$$

$\square$

### A.2 PROOF OF THEOREM 2

Similar to the proof of Theorem 1, there is

$$\begin{aligned} \Delta N &= |Z(\mathcal{D}_k) \cap \Delta B| \\ &= N_k \cdot p_k(C) \cdot \Delta C. \end{aligned} \quad (16)$$

Unlike for the g-SCE, we can exactly calculate $\text{Vol}(\Delta B)$ for the MMC loss. Note that the solution of $\mathcal{L}_{\text{MMC}} = C$ is the hypersphere:

$$\|z - \mu_y^*\|_2^2 = 2C. \quad (17)$$

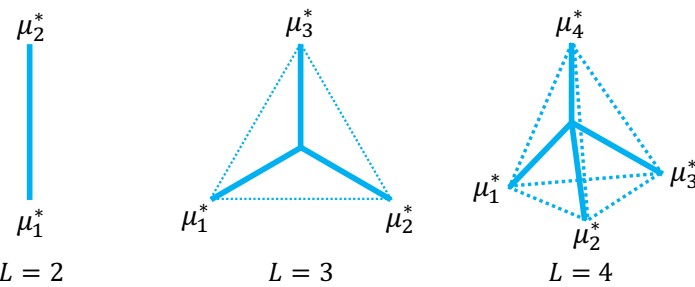

Figure 5: Intuitive illustration of the Max-Mahalanobis centers in the cases of $L = 2, 3, 4$.

According to the formula of the hypersphere surface area (Loskot & Beaulieu, 2007), we have

$$\text{Vol}(\Delta B) = \frac{2^{\frac{d+1}{2}} \pi^{\frac{d}{2}} C^{\frac{d-1}{2}}}{\Gamma(\frac{d}{2})} \cdot \Delta C, \tag{18}$$

where $\Gamma(\cdot)$ is the gamma function. Finally we can obtain the sample density as

$$\begin{aligned} \mathbb{SD}(z) &= \frac{\Delta N}{\Delta B} \\ &\propto \frac{N_k \cdot p_k(C)}{C^{\frac{d-1}{2}}}. \end{aligned} \tag{19}$$

$\square$

# B  TECHNICAL DETAILS

In this section, we provide more technical details we applied in our paper. Most of our experiments are conducted on the NVIDIA DGX-1 server with eight Tesla P100 GPUs.

## B.1  GENERATION ALGORITHM FOR THE MAX-MAHALANOBIS CENTERS

We give the generation algorithm for crafting the Max-Mahalanobis Centers in Algorithm 1, proposed by Pang et al. (2018). Note that there are two minor differences from the originally proposed algorithm. First is that in Pang et al. (2018) they use $C = \|\mu_i\|_2^2$, while we use $C_{\text{MM}} = \|\mu_i\|_2$. Second is that we denote the feature $z \in \mathbb{R}^d$, while they denote $z \in \mathbb{R}^p$. The Max-Mahalanobis centers generated in the low-dimensional cases are quite intuitive and comprehensible as shown in Fig. 5. For examples, when $L = 2$, the Max-Mahalanobis centers are the two vertexes of a line segment; when $L = 3$, they are the three vertexes of an equilateral triangle; when $L = 4$, they are the four vertexes of a regular tetrahedron.

---

**Algorithm 1** GenerateMMcenters

---

**Input:** The constant $C_{\text{MM}}$, the dimension of vectors $d$ and the number of classes $L$. ($L \le d + 1$)
**Initialization:** Let the $L$ mean vectors be $\mu_1^* = e_1$ and $\mu_i^* = 0_d, i \ne 1$. Here $e_1$ and $0_d$ separately denote the first unit basis vector and the zero vector in $\mathbb{R}^d$.
**for** $i = 2$ **to** $L$ **do**
    **for** $j = 1$ **to** $i - 1$ **do**
        $\mu_i^*(j) = -[1 + \langle \mu_i^*, \mu_j^* \rangle \cdot (L-1)]/[\mu_j^*(j) \cdot (L-1)]$
    **end for**
    $\mu_i^*(i) = \sqrt{1 - \|\mu_i^*\|_2^2}$
**end for**
**for** $k = 1$ **to** $L$ **do**
    $\mu_k^* = C_{\text{MM}} \cdot \mu_k^*$
**end for**
**Return:** The optimal mean vectors $\mu_i^*, i \in [L]$.

---

## B.2 WHY THE SQUARED-ERROR FORM IS PREFERRED

In the feature space, penalizing the distance between the features and the prefixed centers can be regarded as a regression problem. In the MMC loss, we apply the squared-error form as $\|z - \mu_y^*\|_2^2$. Other substitutes could be the absolute form $\|z - \mu_y^*\|_2$ or the Huber form. As stated in Friedman et al. (2001), the absolute form and the Huber form are more resistant to the noisy data (outliers) or the misspecification of the class labels, especially in the data mining applications. However, in the classification tasks that we focus on in this paper, the training data is clean and reliable. Thus the squared-error form can lead to high accuracy with faster convergence rate compared to other forms. Furthermore, in the adversarial setting, the adversarial examples have similar properties as the outliers. When we apply the AT mechanism in the training procedure, we expect the classifiers to pay more attention to the adversarial examples, i.e., the outliers. Note that this goal is the opposite of it in the data mining applications, where outliers are intended to be ignored. Therefore, due to the sensitivity to the outliers, the squared-error form can better collaborate with the AT mechanism to improve robustness.

Besides, the MMC loss can naturally perform stronger AT mechanism without additional regularizer term. Specifically, let $x$ be the clean input, $x^*$ be the adversarial example crafted based on $x$, then in the adversarial logit pairing (ALP) method (Kannan et al., 2018), there is an extra regularizer except for SCE as:

$$\|z(x) - z(x^*)\|_2^2. \tag{20}$$

When adding $x^*$ as an extra training point for MMC, then the MMC loss will minimize $\|z(x) - \mu_y^*\|_2^2 + \|z(x^*) - \mu_y^*\|_2^2$, which is an upper bound for $\frac{1}{2}\|z(x) - z(x^*)\|_2^2$. Thus performing naive adversarial training (Goodfellow et al., 2015; Madry et al., 2018) with MMC is equivalent to performing stronger adversarial training variants like ALP. As analyzed above, the squared-error form in the MMC loss can accelerate the convergence of the AT mechanism, since the objective is sensitive to the crafted adversarial examples.

## B.3 VARIANTS OF THE MMC LOSS

In the MMC loss, we encourage the features to gather around the preset Max-Mahalanobis (MM) centers $\mu^* = \{\mu_l^*\}_{l \in [L]}$, which leads to many attractive properties. However, this 'hard' supervision, which induces quite an orderly feature distribution may beyond the reach of the model capability, especially when the classification tasks themselves are already challenging to learn, e.g., ImageNet (Deng et al., 2009). Therefore, we propose potential variants of the MMC loss that could probably solve the problem and make our method more adaptable. We leave the experimental investigations as future work.

Note that the MMC loss can be regarded as minimizing the negative log likelihood (NLL) of $-\log(P(z|y))$, where the conditional feature distribution is modeled as $z|y \sim \mathcal{N}(\mu_y^*, I)$. As described above, this distribution model may not be easy to learn by the DNNs in some cases. Thus, we construct a softer model: $z|y, \mu_y \sim \mathcal{N}(\mu_y, I)$ and $\mu_y \sim \mathcal{N}(\mu_y^*, \alpha I)$, where $\alpha > 0$ is a scalar. Here we give the feature center $\mu_y$ a prior distribution, while the prior is centered at $\mu_y^*$. Intuitively, we relax the constraint that the features have to gather around $\mu_y^*$. Instead, we encourage the features to gather around a substitute $\mu_y$, while $\mu_y$ should be in the vicinity of $\mu_y^*$. In the training, we minimize the joint NLL of $-\log(P(z, \mu_y|y)) = -\log(P(z|y, \mu_y)) - \log(P(\mu_y))$, which is equivalent to minimize the what we call **elastic Max-Mahalanobis center (EMC) loss** as:

$$\mathcal{L}_{\text{EMC}}(Z(x), y) = \frac{1}{2}\|z - \mu_y\|^2 + \frac{1}{2\alpha}\|\mu_y - \mu_y^*\|^2. \tag{21}$$

Here $\mu = \{\mu_l\}_{l \in [L]}$ are simply extra trainable parameters, the prior variance $\alpha$ is a hyperparameter. When $\alpha \to 0$, the EMC loss degenerates to the MMC loss. Note that although $\mu_l^*$ are all on the hypersphere $\{\mathbf{z} \in \mathbb{R}^d | \|\mathbf{z}\| = C_{\text{MM}}\}$, the support sets of $\mu_l$ are the entire feature space $\mathbb{R}^d$.

Further improvement can be made w.r.t. the MM centers $\mu^*$. An implicit assumption behind the generation process of $\mu^*$ is that any two classes are mutually independent. This assumption could be approximately true for MNIST and CIFAR-10, but for more complex datasets, e.g., CIFAR-100 or ImageNet, this assumption may not be appropriate since there are structures in the relation among classes. These structures can usually be visualized by a tree. To solve this problem, we introduce the **hierarchical Max-Mahalanobis (HM) centers** $\mu^H = \{\mu_l^H\}_{l \in [L]}$, which adaptively craft the centers

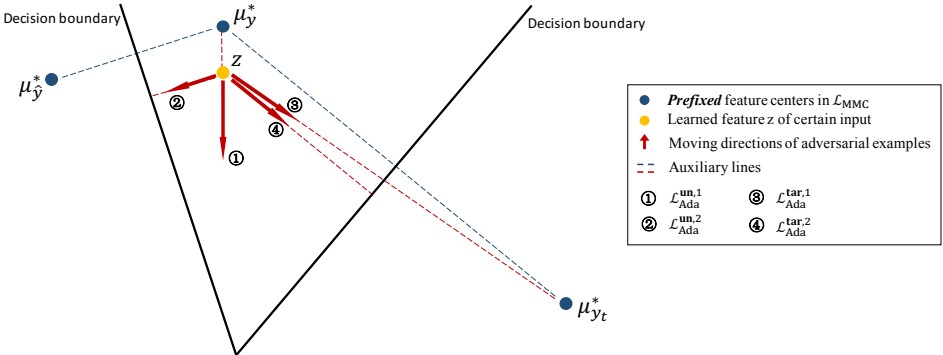

Figure 6: Intuitive demonstration of the attacking mechanisms under different adaptive objectives. Here $y$ is the original label, $\tilde{y} = \arg\max_{l \neq y} h_l$ is the label of the nearest other decision region w.r.t. the feature $z$, and $y_t$ is the target label of targeted attacks.

according to the tree structure. Specifically, we first assign a virtual center (i.e., the origin) to the root node. For any child node $n_c$ in the tree, we denote its parent node as $n_p$, and the number of its brother nodes as $L_c$. We locally generate a set of MM centers as $\mu^{(s,L_c)} = \text{GenerateMMcenters}(C^s, d, L_c)$, where $s$ is the depth of the child node $n_c$, $C^s$ is a constant with smaller values for larger $s$. Then we assign the virtual center to each child node of $n_p$ from $\mu_{n_p} + \mu^{(s,L_c)}$, i.e., a shifted set of crafted MM centers, where $\mu_{n_p}$ is the virtual center assigned to $n_p$. If the child node $n_c$ is a leaf node, i.e., it correspond to a class label $l$, then there is $\mu_l^H = \mu_{n_c}$. For example, in the CIFAR-100 dataset, there are 20 superclasses, with 5 classes in each superclass. We first craft 20 MM centers as $\mu^{(1,20)} = \text{GenerateMMcenters}(C^1, d, 20)$ and 5 MM centers as $\mu^{(2,5)} = \text{GenerateMMcenters}(C^2, d, 5)$, where $C^2 \ll C^1$. Note that $\mu^{(2,5)}$ could be different for each superclass, e.g., by a rotation transformation. Then if the label $l$ is the $j$-th class in the $i$-th superclass, there is $\mu_l^H = \mu_i^{(1,20)} + \mu_j^{(2,5)}$.

## B.4 ADAPTIVE OBJECTIVES AND THE INDUCED ATTACKING MECHANISMS

We apply the adaptive versions of existing attacks when evading the networks trained by the MMC loss. We separately design two adaptive adversarial objectives $\mathcal{L}_{\text{Ada}}$ to minimize under the untargeted mode: $\mathcal{L}_{\text{Ada}}^{\mathbf{un},1} = -\mathcal{L}_{\text{MMC}}(z,y)$; $\mathcal{L}_{\text{Ada}}^{\mathbf{un},2} = \mathcal{L}_{\text{MMC}}(z,\tilde{y}) - \mathcal{L}_{\text{MMC}}(z,y)$, and under the targeted mode: $\mathcal{L}_{\text{Ada}}^{\mathbf{tar},1} = \mathcal{L}_{\text{MMC}}(z,y_t)$; $\mathcal{L}_{\text{Ada}}^{\mathbf{tar},2} = \mathcal{L}_{\text{MMC}}(z,y_t) - \mathcal{L}_{\text{MMC}}(z,y)$, where $y_t$ is the targeted label, $\tilde{y}$ is generally the highest predicted label except for $y$ as defined in Sec. 3.2. These objectives refer to previous work by Carlini & Wagner (2017a;b). Specifically, the adaptive objectives $\mathcal{L}_{\text{Ada}}^{\mathbf{tar},1}$ and $\mathcal{L}_{\text{Ada}}^{\mathbf{un},1}$ are used in the PGD, MIM and SPSA attacks, while the objectives $\mathcal{L}_{\text{Ada}}^{\mathbf{tar},2}$ and $\mathcal{L}_{\text{Ada}}^{\mathbf{un},2}$ are used in the C&W attacks.

In Fig. 6, we demonstrate the attacking mechanisms induced by different adaptive adversarial objectives. Note that we only focus on the gradients and ignore the specific method which implements the attack. Different adaptive objectives are preferred under different adversarial goals. For examples, when decreasing the confidence of the true label is the goal, $\mathcal{L}_{\text{Ada}}^{\mathbf{un},1}$ is the optimal choice; in order to mislead the classifier to predict an untrue label or the target label, $\mathcal{L}_{\text{Ada}}^{\mathbf{un},2}$ and $\mathcal{L}_{\text{Ada}}^{\mathbf{tar},2}$ are the optimal choices, respectively. Sometimes there are additional detectors, then the adversarial examples generated by $\mathcal{L}_{\text{Ada}}^{\mathbf{tar},1}$ could be assigned to the target label with high confidence by the classifiers.

## B.5 RELATED WORK IN THE FACE RECOGNITION AREA

There are many previous work in the face recognition area that focus on angular margin-based softmax (AMS) losses (Liu et al., 2016; 2017; Liang et al., 2017; Wang et al., 2018; Deng et al., 2019). They mainly exploit three basic operations: weight normalization (WN), feature normalization (FN), and angular margin (AN). It has been empirically shown that WN can benefit the cases with unbalanced data (Guo & Zhang, 2017); FN can encourage the models to focus more on hard examples (Wang et al., 2017); AN can induce larger inter-class margins and lead to better generalization in different facial tasks (Wang et al., 2018; Deng et al., 2019). However, there are two critical differences between our MMC loss and these AMS losses:

Table 5: Classification accuracy (%) on the *white-box* adversarial examples crafted on the test set of **CIFAR-10** and **CIFAR-100**. The results w.r.t the MMC loss are reported under the adaptive versions of different attacks. MMC can better exploit deep architectures, while SCE cannot.

| Methods | Cle. | Perturbation $\epsilon = 8/255$ | | | | Perturbation $\epsilon = 16/255$ | | | |
|---|---|---|---|---|---|---|---|---|---|
| | | $\mathrm{PGD}_{10}^{\mathbf{tar}}$ | $\mathrm{PGD}_{10}^{\mathbf{un}}$ | $\mathrm{PGD}_{50}^{\mathbf{tar}}$ | $\mathrm{PGD}_{50}^{\mathbf{un}}$ | $\mathrm{PGD}_{10}^{\mathbf{tar}}$ | $\mathrm{PGD}_{10}^{\mathbf{un}}$ | $\mathrm{PGD}_{50}^{\mathbf{tar}}$ | $\mathrm{PGD}_{50}^{\mathbf{un}}$ |
| **CIFAR-10** | | | | | | | | | |
| SCE (Res.32) | 93.6 | $\leq 1$ | 3.7 | $\leq 1$ | 3.6 | $\leq 1$ | 2.7 | $\leq 1$ | 2.9 |
| MMC (Res.32) | 92.7 | **48.7** | **36.0** | **26.6** | **24.8** | **36.1** | **25.2** | **13.4** | **17.5** |
| SCE (Res.110) | 94.7 | $\leq 1$ | 3.0 | $\leq 1$ | 2.9 | $\leq 1$ | 2.1 | $\leq 1$ | 2.0 |
| MMC (Res.110) | 93.6 | **54.7** | **46.0** | **34.4** | **31.4** | **41.0** | **30.7** | **16.2** | **21.6** |
| **CIFAR-100** | | | | | | | | | |
| SCE (Res.32) | 72.3 | $\leq 1$ | 7.8 | $\leq 1$ | 7.4 | $\leq 1$ | 4.8 | $\leq 1$ | 4.7 |
| MMC (Res.32) | 71.9 | **23.9** | **23.4** | **15.1** | **21.9** | **16.4** | **16.7** | **8.0** | **15.7** |
| SCE (Res.110) | 74.8 | $\leq 1$ | 7.5 | $\leq 1$ | 7.3 | $\leq 1$ | 4.7 | $\leq 1$ | 4.5 |
| MMC (Res.110) | 73.2 | **34.6** | **22.4** | **23.7** | **16.5** | **24.1** | **14.9** | **13.9** | **10.5** |

**Difference one: The inter-class margin**

- The AMS losses induce the inter-class margins mainly by encouraging the intra-class compactness, while the weights are not explicitly forced to have large margins (Qi & Zhang, 2018).

- The MMC loss simultaneously fixes the class centers to be optimally dispersed and encourages the intra-class distribution to be compact. Note that both of the two mechanisms can induce inter-class margins, which can finally lead to larger inter-class margins compared to the AMS losses.

**Difference two: The normalization**

- The AMS losses use both WN and FN to exploit the angular metric, which makes the normalized features distribute on hyperspheres. The good properties of the AMS losses are at the cost of abandoning the radial degree of freedom, which may reduce the capability of models.

- In the MMC loss, there is only WN on the class centers, i.e., $\|\mu_y^*\| = C_{\mathrm{MM}}$, and we leave the degree of freedom in the radial direction for the features to keep model capacity. However, note that the MMC loss $\|z - \mu_y^*\|_2^2 \geq (\|z\|_2 - C_{\mathrm{MM}})^2$ is a natural penalty term on the feature norm, which encourage $\|z\|_2$ to not be far from $C_{\mathrm{MM}}$. This prevents models from increasing feature norms for easy examples and ignoring hard examples, just similar to the effect caused by FN but more flexible.

