# OpenReview forum: "Rethinking Softmax Cross-Entropy Loss for Adversarial Robustness"
_ICLR.cc/2020/Conference — Accept (Poster)_

### Official Review · AnonReviewer1 · 2019-10-22
**Official Blind Review #1**

**Rating:** 6

**Review:**

[Post rebuttal: The rebuttal addresses most of my concerns. I have updated my score.]

This paper points out some limitation of the traditional softmax cross-entropy (SCE) loss, e.g., causing indirect and unexpected supervisory signals on the learned features so that the points with low loss values tend to spread over the space sparsely. To remedy this, the authors then proposes the max-mahalanobis center (MMC) loss. By analyzing the sample density, it is proved that the MMC loss has a sample density proportional to the number of data samples in each class. This means MMC loss induces higher feature densities than the SCE loss, thus is expected to be more robust under adverssarial attacks.

I find the paper fairly well written in general, and the arguments are well supported by the development of the theoretical results. There are, however, several questions raised by the reviewer.

One question is that by looking at the definition (8), it seems the MMC loss essentially tries to move data from different classes apart. This seems to be quite similar to the max margin loss. I would expect the authors differentiate the differences between these two. Consequently, I would also like to see empirical comparisons between these two losses.

Another question is even if you have Theorem 2, the transfer from Theorem 2 to robustness does not seem direct, e.g., how do you guarantee when the density form in eq.9 is better than that in eq.7 for robustness?

Then when looking back at the sample density definition in eq. (2). I wonder how the Vol is defined to guarantee eq.2 is a valid density function? e.g., to guarantee the integration equals to 1.

In terms of experiments, since it is claimed that the proposed loss adds little computation cost compared to the SCE loss, I think it is better to include running time comparison in the results.

The last two lines in page 8, what are unseen attacks? From the experiments, it seems that it means the PGD with different steps. I think the authors  should be careful to use the term, because some literature use the term *different attacks* to refer to attacks with different *norms*.

Second line below Table 5: it is said the MMC loss keeps state-of-the-art performance on clean data. However, by looking at Table 4, the state-of-the-art is obvious SCE. Why did you say that?

Others:
1. S_k,\tilde{k}^2 is undefined.
2. How do you solve the minmax problem below eq.8? I guess you use the algorithm in Appendix B.1? Please clarify.


**Experience Assessment:**

I have published one or two papers in this area.

**Review Assessment: Checking Correctness Of Derivations And Theory:**

I assessed the sensibility of the derivations and theory.

**Review Assessment: Checking Correctness Of Experiments:**

I assessed the sensibility of the experiments.

**Review Assessment: Thoroughness In Paper Reading:**

I read the paper at least twice and used my best judgement in assessing the paper.

---

> ### Author Response · Authors · 2019-11-06
> **Thank you for your valuable review**
>
> Thank you for your valuable review.
>
> Question 1: Difference between max-margin loss and MMC:
> Generally, MMC loss can encourage intra-class compact (higher sample density) in training, while keeping inter-class dispersion by the MM centers. In comparison, max-margin loss like SVM can only induce inter-class dispersion. Besides, as shown in our experiments and other empirical results (https://github.com/adversarial-robustness-benchmark/adversarial-robustness-benchmark ), only inducing inter-class large margin like MMLDA, L-GM or DeepDefense [1] cannot provide reliable robustness under adaptive attacks.
>
> [1] Yan et al. Deep defense: Training dnns with improved adversarial robustness. NeurIPS 2018.
>
>
> Question 2: About the conclusion in Theorem 2:
> We do not intend to claim that higher sample density is a sufficient condition for robustness. Instead, as indicated by Schmidt et al. (2018), having enough samples is generally a necessary condition to train robust models, both globally (sample complexity) and locally (sample density). So the density form in Eq. (9) is better than Eq. (7) indicates that MMC loss can better achieve the necessary condition for robust learning, given a fixed amount of training data. Then in our experiments, we further empirically verify that the higher density induced by MMC can indeed lead to better robustness.
>
>
> Question 3: Definition of sample density:
> The sample density in Eq. (2) denotes 'number of training samples per unit volume', so the integration over whole space is the number of training samples in the dataset $D$, i.e., $N=|D|$. This can be formally denoted as $\int \text{SD}\cdot d V=\int d N=N$.
>
>
> Question 4: Computational cost of MMC:
> We perform our experiments on a single NVIDIA Tesla P100 GPU. Both SCE and MMC are trained by 200 epochs, as shown in Figure 3(a). The training time per epoch is
>                                  SCE         MMC
> Resnet-32   ||  48~51s   ||  49~51s   ||
> Resnet-110  || 189~211s || 192~214s  ||
>
> Besides, we indeed use the algorithm in Appendix B.1 to solve the min-max problem Eq.(8), we will make this clearer in the revision. We implement the algorithm in Matlab code, and it takes less than one second to craft the MM centers used in this paper.
>
>
> Question 5: About the terms:
> Thank you for pointing out. We use PGD-10 with $\epsilon=8/255$ in AT, so the 'unseen attacks' refer to those with different steps (e.g., PGD-50) or different norms (e.g., $\epsilon=16/255$) as shown in Table 1.
>
> We upload a revision, the modifications include:
> 1. We change the claim of 'state of the art' to 'comparable' to avoid an overclaim as you suggested;
>
> 2. We remove the term 'unseen attacks' and directly refer to the attacks different from those used in AT;
>
> 3. We clarify the algorithm used to solve the minmax problem below Eq.(8);
>
> 4. We remove the Gaussian assumption in Theorem 1 and  Theorem 2 to get a more general form.

---

> > ### Comment · AnonReviewer1 · 2019-11-15
> > **thanks for the clarification and revision**
> >
> > Thanks for the clarification and revision made, which solve most of my concerns.
> >
> > There is, however, one more question: It seems the loss by Wen et al. (2016) is very similar to yours, thus should be compared with. Although you refer the paper in the experiment, I am not quite sure whether you have compared with their method? I saw MMLDA, is this algorithm their method (it is referred to with multiple papers)?

---

> > > ### Author Response · Authors · 2019-11-15
> > > **Thank you for your feedback!**
> > >
> > > The method 'Center loss' is exactly the method of Wen et al. (2016). You can find it in, e.g., Table 1, Table 2, Table 4 and Figure 3.
> > >
> > > MMLDA is the method of Pang et al. (2018), as detailed in Remark 4.
> > >
> > > Pang et al.  Max-mahalanobis linear discriminant analysis networks. ICML 2018
> > >
> > >
> > > Since it is almost the deadline of rebuttal, we will keep online for your further feedback, thank you!

---

> > > > ### Comment · AnonReviewer1 · 2019-11-15
> > > > **thanks**
> > > >
> > > > OK, I think it is better to clarify this at the beginning of the experiment section.
> > > >
> > > > Overall, I am satisfied with the revision, and will update my score.

---

> > > > > ### Author Response · Authors · 2019-11-15
> > > > > **Thank you!**
> > > > >
> > > > > Thank you for the suggestion, we have uploaded a new revision to clarify this clearer at the beginning of the experiment section.

---

### Official Review · AnonReviewer3 · 2019-10-22
**Official Blind Review #3**

**Rating:** 6

**Review:**

The paper compares between SCE loss,  large-margin Gaussian Mixture (L-GM) loss and proposes the Max-Mahalanobis center (MMC) loss as an alternative to explicitly learn more structured representations and induce high-density regions in the feature space. Overall the paper is well written, with sufficient theoretical reasoning and experiments. However, the reviewer has the following concerns and questions,
The theoretical analysis depends largely on the Gaussian assumption and argues that when the loss is distributed as Gaussian, it seems to be not even a fair comparison since assuming L_{MMC} is gaussian is totally different from assuming L_{g-SCE} is Gaussian. Also in practice it is hard to justify whether certain loss function really behaves like a Gaussian distribution, which makes the application of the theorem more limited. In fact, if the samples are concentrated (which can be common in practice), is the proposed method still able to induce high density sample region?
The experiments give very competitive results for MMC loss. It would also be interesting to see if implementing other defenses or do an adversarial training would still make MMC loss much better than other loss (at least from the AT example, it seems that MMC does not perform uniformly better than SCE as before).
Are the experiment results sensitive to the choice of parameters C_MM and L?

I have read the author responses and I think they are quite solid. I have updated my score.


**Experience Assessment:**

I have read many papers in this area.

**Review Assessment: Checking Correctness Of Derivations And Theory:**

I assessed the sensibility of the derivations and theory.

**Review Assessment: Checking Correctness Of Experiments:**

I assessed the sensibility of the experiments.

**Review Assessment: Thoroughness In Paper Reading:**

I read the paper at least twice and used my best judgement in assessing the paper.

---

> ### Author Response · Authors · 2019-11-06
> **Thank you for your valuable review**
>
> Thank you for your valuable review.
>
> Question 1. Gaussian assumption:
> We upload a revision, where we remove the Gaussian assumption in Theorem 1 and  Theorem 2 to get a more general form. In experiments, we find that $p_{k}(c)$ in MMC has much higher probability nearby the loss $c=0$, comparted to $p_{k,\tilde{k}}(c)$ in SCE, which further leads to higher sample density beyond the theoretical analyses.
>
> Besides, when we analyze the conclusions of Theorem 1 and Theorem 2 in the original version, we also do not depend on any of the Gaussian parameters like $C_{k}$ / $C_{k,\tilde{k}}$, $S_{k}$ / $S_{k,\tilde{k}}$ or even the Gaussian density function $\varphi(x)$. We demonstrate the superiority of MMC over SCE based on the comparison between $N_{k}$ and $N_{k,\tilde{k}}$, and the denominator terms in Eq. (7) and Eq. (9) that are independent of the Gaussian assumption.
>
> Actually, the only use of the Gaussian assumption is to make the representation of sample density more specific in Theorem 1 and Theorem 2. This assumption can be removed to obtain a more generalized formula of sample density without changing our analyses and conclusions, just as shown in the revision.
>
>
> Question 2: Results on AT:
> The improvement of combining MMC with AT is mainly in the cases when models are evaded by the attacks different from those used in training. For example, in Table 1, MMC + AT (tar) can have 45.1% accuracy under PGD-50 (un, $\epsilon=16/255$) attack, while SCE + AT (tar) can only have 16.0% accuracy.
>
>
> Question 3: Sensitivity to the choice of $C_{\text{MM}}$ and $L$:
> We have done the experiments of $C_{\text{MM}}=100$ and compare it with the results of MMC-10 in Table 1, as shown below:
>
>    Method ||Clean || P-10(tar) | P-10(un) | P-50(tar) | P-50(un) || P-10(tar) | P-10(un) | P-50(tar) | P-50(un) ||
> MMC-10  || 92.7 || 48.7 | 36.0 | 26.6 | 24.8 || 36.1 | 25.2 | 13.4 | 17.5 ||
> MMC-100 || 92.8 || 48.4 | 35.6 | 26.4 | 24.3 || 35.8 | 25.0 | 13.1 | 17.3 ||
>
> We can see that MMC-100 has similar performance as MMC-10. In Table 4 and Table 5, we show the effectiveness of our method on CIFAR-100, where $L=100$. So the performance of MMC loss is relatively not sensitive to the choice of $C_{\text{MM}}$ and $L$.

---

> ### Author Response · Authors · 2019-11-14
> **Thank you for updating the score**
>
> We really appreciate it!

---

### Official Review · AnonReviewer2 · 2019-10-29
**Official Blind Review #2**

**Rating:** 6

**Review:**

This paper first shows some potential issues of softmax loss (i.e., cross-entropy loss with softmax function) and then propose the Max-Mahalanobis center (MMC) loss to encourge the intra-class compactness for better adversarial robustness.

The MMC loss is essentially minimizing the distance between the feature and the pre-fixed class center. Different from center loss, these centers are determined by minimizing the maximum inner product between any two class centers. Since the norm of these class centers are normalized to a constant. It is equivalent to angles. This acutally reminds me of a number of works in angular margin-based softmax loss. Just to name a few:

[1] Large-Margin Softmax Loss for Convolutional Neural Networks, ICML 2016
[2] SphereFace: Deep Hypersphere Embedding for Face Recognition, CVPR 2017
[3] Soft-margin softmax for deep classification, ICNIP 2017
[4] CosFace: Large Margin Cosine Loss for Deep Face Recognition, CVPR 2018
[5] ArcFace: Additive Angular Margin Loss for Deep Face Recognition, CVPR 2019

I think these works are closely related to what the authors aim to do, and therefore they should be discussed methodologically and compared empirically.

Besides that, I think it is also worth conducting an ablation study for how to determine these class centers. This paper considers to minimize the maximum inner product. There are a few papers listed below that explicitly discusses how to make the class centers uniformly spaced. The authors may consider to compare these methods for determining the class centers.

[1] Learning towards Minimum Hyperspherical Energy, NeurIPS 2018
[2] UniformFace: Learning Deep Equidistributed Representation for Face Recognition, CVPR 2019

For the experiments, the MMC loss indeed shows some advantages over the softmax loss. I am basically convinced by the experiments, although it can further strengthen the paper if the authors can conduct some evaluations on large-scale datasets like ImageNet.

I appreciate the authors provide many theoretical justifications, which is inspiring. Intuitively speaking, I can understand that shrinking the feature space (i.e., make feature distribution more compact) can improve the adversarial robustness. As a result, I think this paper is naturally motivated and is also theoretically sound. The experiments can be further improved.

**Experience Assessment:**

I have read many papers in this area.

**Review Assessment: Checking Correctness Of Derivations And Theory:**

I assessed the sensibility of the derivations and theory.

**Review Assessment: Checking Correctness Of Experiments:**

I assessed the sensibility of the experiments.

**Review Assessment: Thoroughness In Paper Reading:**

I read the paper at least twice and used my best judgement in assessing the paper.

---

> ### Author Response · Authors · 2019-11-06
> **Thank you for the supportive review**
>
> Thank you for the supportive review.
>
> We upload a revision which includes discussion on related angular margin-based softmax (AMS) losses in Appendix B.5 (due to limited space in the main text). We highlight the two main differences between MMC and AMS  losses below:
>
> Difference one: The inter-class margin
>
> The AMS losses induce the inter-class margins mainly by encouraging the intra-class compactness, while the weights are not explicitly forced to have large margins.
>
> The MMC loss simultaneously fixes the class centers to be optimally dispersed and encourages the intra-class distribution to be compact. Note that both of the two mechanisms can induce inter-class margins, which can finally lead to larger inter-class margins compared to the AMS losses.
>
>
> Difference two: The normalization
>
> The AMS losses use both weight normalization (WN) and feature normalization (FN) to exploit the angular metric, which makes the normalized features distribute on hyperspheres. The good properties of the AMS losses are at the cost of abandoning the radial degree of freedom, which may reduce the capability of models.
>
> In the MMC loss, there is only WN on the class centers, i.e., $\|\mu_{y}^*\|=C_{\text{MM}}$, and we leave the degree of freedom in the radial direction for the features to keep model capacity. However, note that the MMC loss $\|z-\mu_{y}^*\|_{2}^2\geq (\|z\|_{2}-C_{\text{MM}})^2$ is a natural penalty term on the feature norm, which encourage $\|z\|_{2}$ to not be far from $C_{\text{MM}}$. This prevents models from increasing feature norms for easy examples and ignoring hard examples, just similar to the effect caused by FN but more flexible.
>
>
> As to the class centers, an ablation study on uniformly centers is a good suggestion. We will check the mentioned work and conduct some necessary experiments.

---

> ### Author Response · Authors · 2019-11-11
> **Update on ablation study experiments**
>
> We have uploaded a newer revision, which includes the ablation study in Table 1 and described in Section 4.3.
>
> To investigate the effect on robustness induced by high sample density in MMC, we substitute uniformly sampled center set $\mu^{r}=\{\mu_{l}^{r}\}_{l\in[L]}$ for the MM center set $\mu^{*}$, and name the resulted method as "MMC-10 (rand)". There is also $\|\mu_{l}^{r}\|_{2}=C_{\text{MM}}$, but $\mu^{r}$ is no longer the solution of the min-max problem in Section 3.3.
>
> From the results in Table 1, we can see that higher sample density alone in "MMC-10 (rand)" can already lead to much better robustness than other baseline methods even under the adaptive attacks, while using the optimal center set $\mu^*$ as in "MMC-10" can further improve performance.

---

> > ### Comment · AnonReviewer2 · 2019-11-15
> > **Satisfied Rebuttal**
> >
> > Thanks for the detailed clarification. I am satisfied with the author's rebuttal.
> >
> > I will rate this paper as "accept" in normal cases, but since we should apply a rigorous judgement for 10-page paper, I am sitting between "weak accept" and "accept".

---

### Official Review · AnonReviewer4 · 2019-11-01
**Official Blind Review #4**

**Rating:** 8

**Review:**

This paper proposes an alternative loss function for classification models that they claim leads to improved adversarial robustness even under strong adaptive attacks. It also attempts to analyze how the softmax cross-entropy loss discourages robustness by considering the problem in terms of local density in the pre-logit feature space.

Although as an emergency reviewer I haven't had time for a thorough verification, the overall idea seems sound, as do their theoretical and experimental results. I appreciate the careful analysis of the sample densities induced by each method; the N/L^2 vs. N/L result is especially nice. They also seem to follow best practices for evaluating adversarial defenses. Overall I do feel like the research topic of developing alternate loss functions and identifying pathologies with current popular loss functions is important and maybe insufficiently studied / publicized, so I think publishing this paper would be helpful for the field.

A few questions:
(1) If the MMC loss makes your final models so much more robust to small perturbations, why is there still such a large clean accuracy drop when combining your method with adversarial training in Figure 3a? I would have hoped that the tradeoff would have been less extreme. If you started adversarial training midway through the training process, or used a smaller perturbation size, would the tradeoff still be as large as with SCE models?
(2) It makes sense that the optimal method of choosing MM centers would be to place them at the vertices of simplexes when #dims = #classes, but if there was some other way of avoiding the degradation problem from Wen et al. 2016, would it ever make sense (especially when #dims < #classes) to allow some automatic slackening of the MM centers (i.e. allow them to move from their original positions, but at a heavy cost), in order to permit classes that are similar (e.g. different breeds of imagenet dogs) to cluster more closely together, or perhaps to allow for classes with different levels of dispersion? Something feels suboptimal about forcing class centers to be equidistant and identical. This isn't intended as a major criticism but it might be an interesting direction for future work.

**Experience Assessment:**

I have published one or two papers in this area.

**Review Assessment: Checking Correctness Of Derivations And Theory:**

I did not assess the derivations or theory.

**Review Assessment: Checking Correctness Of Experiments:**

I assessed the sensibility of the experiments.

**Review Assessment: Thoroughness In Paper Reading:**

I made a quick assessment of this paper.

---

> ### Author Response · Authors · 2019-11-06
> **Thank you for the supportive review**
>
> Thank you for the supportive review.
>
> Question 1. The tradeoff between clean and adversarial accuracy:
> Thank you for the insightful suggestions, which are enlightening for us. We will try these techniques to improve the tradeoff.
>
>
> Question 2. More flexible variants:
> Actually, we have proposed two flexible variants of MMC loss in Appendix B.3. One is called elastic Max-Mahalanobis center (EMC) loss, which regards MMC centers as a prior and adds a penalty term to slacken the real position of centers, just as you suggested. Another one is called hierarchical Max-Mahalanobis (HM) loss, which uses a hierarchical center set to deal with the more complicated datasets.

---

> > ### Comment · AnonReviewer4 · 2019-11-13
> > **Thanks for your response**
> >
> > Apologies for missing those variants of the MMC loss in the paper -- those make total sense! On reading your responses both to my review and the others, I'm inclined to slightly strengthen my acceptance rating -- I think you all did a very good job responding to the criticisms other reviewers raised, and it's worth emphasizing that all reviewers seem to agree this paper is well-written with sufficient experiments (which isn't true of many papers in this subfield). I hope the other reviewers will consider improving their scores in light of your thorough response.

---

> > > ### Author Response · Authors · 2019-11-14
> > > **Thank you!**
> > >
> > > Thank you! We really appreciate the support and approval.

---

### Author Response · Authors · 2019-11-11
**Looking forward to further feedbacks**

Dear Reviewers,

Thank you again for your valuable comments and suggestions, which are really helpful for us. We have uploaded new revisions and posted responses to the proposed concerns and questions.

We totally understand that this is a quite busy period of time, since the reviewers may be preparing the rebuttal for their own submissions or rushing for the deadline of the recent conferences.

So we deeply appreciate it if the reviewers can take some time to return further feedbacks on whether our responses and extra experiment results solve their concerns. If there is any other question, we will try our best to provide satisfactory answers.

Best,
The authors

---

### Decision · Program_Chairs · 2019-12-19

**Decision:**

Accept (Poster)

**Comment:**

This paper proposes an alternative loss function, the max-mahalanobis center loss, that is claimed to improve adversarial robustness.

In terms of quality, the reviewers commented on the convincing experiments and theoretical results, and were happy to see the sample density analysis.

In terms of clarity, the reviewers commented that the paper is well-written.

The problem of adversarial robustness is relevant to the ICLR community, and the proposed approach is a novel and significant contribution in this area.

The authors have also convincingly answered the questions of the authors and even provided new theoretical and experimental results in their final upload.